# Synthesis, Crystal Structure, and Biological Evaluation of Novel 5-Hydroxymethylpyrimidines

**DOI:** 10.3390/ma14226916

**Published:** 2021-11-16

**Authors:** Marcin Stolarczyk, Agnieszka Matera-Witkiewicz, Aleksandra Wolska, Magdalena Krupińska, Aleksandra Mikołajczyk, Anna Pyra, Iwona Bryndal

**Affiliations:** 1Department of Organic Chemistry and Drug Technology, Faculty of Pharmacy, Wroclaw Medical University, 211A Borowska, 50-556 Wroclaw, Poland; marcin.stolarczyk@umw.edu.pl (M.S.); iwona.bryndal@umw.edu.pl (I.B.); 2Screening Biological Activity Assays and Collection of Biological Material Laboratory, Wroclaw Medical University, 211A Borowska, 50-556 Wroclaw, Poland; aleksandra.wolska@umw.edu.pl (A.W.); magdalena.krupinska@umw.edu.pl (M.K.); aleksandra.mikolajczyk@umw.edu.pl (A.M.); 3Faculty of Chemistry, University of Wroclaw, Joliot-Curie Street 14, 50-383 Wroclaw, Poland; anna.pyra@chem.uni.wroc.pl

**Keywords:** pyrimidines, anticancer activity, single-crystal X-ray diffraction

## Abstract

Pyrimidine displays a wide array of bioactivities, and thence, it is still considered a potent unit of new drug research. Its derivative, 5-hydroxymethylpyrimidine, can be found as a scaffold of nontypical nitrogen bases in DNA and as a core of some natural bioactive compounds. In this study, we obtained a series of 5-hydroxymethylpyrimidines that vary in the 4-position by the reduction of proper esters. All compounds were characterized by spectroscopic analysis, and single-crystal X-ray diffraction was performed for some of them. Biological investigations estimated cytotoxic properties against normal (RPTEC) and cancer (HeLa, HepaRG, Caco-2, AGS, A172) cell lines. It was found that the derivatives with an aliphatic amino group at the 4-position are generally less toxic to normal cells than those with a benzylsulfanyl group. Moreover, compounds with bulky constituents exhibit better anticancer properties, though at a moderate level. The specific compounds were chosen due to their most promising IC50 concentration for in silico study. Furthermore, antimicrobial activity tests were performed against six strains of bacteria and one fungus. They demonstrated that only derivatives with at least three carbon chain amino groups at the 4-position have weak antibacterial properties, and only the derivative with 4-benzylsulfanyl constituent exhibits any antifungal action.

## 1. Introduction

The pyrimidine ring is a well-known and established constituent of many synthetic drugs commonly used in medicine that demonstrate a variety of pharmacological activities, especially antimicrobial (e.g., trimethoprim, pyrimethamine, zidovudine, flucytosine) and antineoplastic (e.g., fluorouracil, gemcitabine, cytarabine) [1]. It is still among the leading compounds investigated by medicinal chemists for practical clinical applications of its derivatives [2]. It is also ubiquitous in nature due to the presence of nitrogenous bases in DNA and RNA.

The vast majority of pyrimidines present in nucleic acids are cytosine, thymine, and uracil. Other pyrimidine bases occur rarely and constitute only a small fraction of nucleobases [3]. Modifications in the structures of nucleobases include the process of methylation at the 5-position of the pyrimidine ring and further oxidation of this group by TET (ten-eleven translocation) family proteins [4] to form 5-hydroxymethylcytosine (hmC) (Figure 1a) [5] or 5-hydroxymethyluracil (5-hmU) (Figure 1b) [6]. The function of the mentioned nucleobases is still unclear, but they are thought to regulate gene expression or prompt DNA demethylation [7]. The role of TET proteins in cancer formation was also noticed [8], and TET1 was highlighted as a promising target for the treatment of therapy-resistant cancer [9].

The 5-hydroxymethylpyrimidine scaffold can also be found in a naturally occurring antibiotic, bacimethrin (Figure 1c), isolated from *Bacillus megaterium* [10]. The structural resemblance between bacimethrin and toxopyrimidine (Figure 1d), which is the pyrimidine part of vitamin B_1_ (thiamine), lies behind its mechanism of action that involves its conversion to 2′-methoxy-thiamin pyrophosphate. It may act as a thiamin-pyrophosphate-dependent enzyme or gene inhibitor [11]. The antibiotic and its analogs exhibit antibacterial and anticancer properties [12].

In our previous publications, we reported that 5-hydroxymethylpyrimidine with a tetrasulfide bridge at the 4-position has interesting antibacterial and antifungal properties. Its MIC ranged from 4 to 32 mg/mL, depending on the strain [13]. Additionally, we observed that the hydroxylation of 4-[(4-chlorobenzyl)sulfanyl]-5,6-dimethyl-2-phenylpyrimidine to its 5-hydroxymethyl derivative enhances the cytotoxicity significantly towards both normal (survival of normal human endothelial cells, 58% vs. 1%) and cancer cell lines (IC50 in the range > 100–55 μM vs. 17–38 μM, depending on the cancer cell line) [14].

In the continuation of our foregoing research, we present the results of the cytotoxic and antimicrobial study of a series of 5-hydroxymethylpyrimidines that vary in the 4-position and a comparison of their structural properties and biological activity.

## 2. Materials and Methods

### 2.1. Synthesis

Reagents were purchased and used without purification. Amines; benzyl chlorides; LiAlH_4_ (lithium aluminum hydride), reagent grade 95; POCl_3_; silica gel, 200–400 mesh, 60 Å for column chromatography; and solvents for NMR spectroscopy were supplied by Merck, Darmstadt, Germany. Other reagents were provided by Chempur, Piekary Śląskie, Poland. TLC sheets, Alugram SIL G/UV254, were obtained from Macherey-Nagel, Düren, Germany.

All melting points were uncorrected and determined by the open capillary method with an Electrothermal IA9100 melting point apparatus. NMR spectra were recorded using a Bruker ARX 300 MHz NMR spectrometer. The abbreviations used in NMR spectra are: s—singlet, d—doublet, t—triplet, q—quartet, and m—multiplet. IR spectra were recorded with a Thermo Scientific USA Nicolet iS50 FTIR using the ATR technique. MS spectra were recorded with a Bruker Daltonic Compact using the ESI technique.

#### 2.1.1. General Procedure for the Preparation of **2a** and **2b**

Ethyl 4-methyl-2-phenyl-6-sulfanylpyrimidine-5-carboxylate (**1a**) (2.74 g, 10 mmol) [15] was dissolved in a mixture of methanol (20 mL) and NaOH (0.40 g, 10 mmol), and proper benzyl chloride was added (11 mmol). The reaction mixture was stirred with a magnetic stirrer for 3 h at ambient temperature. Then the white solid was filtered off and washed with methanol (10 mL) and water (10 mL) and then dried at atmospheric pressure. Its purity was monitored by TLC using chloroform as eluent.

*Ethyl 4-(benzylsulfanyl)-6-methyl-2-phenylpyrimidine-5-carboxylate* (**2a**). Product characterization: yield, 3.03 g, 83.24%; white solid; melting point, 100–101 °C; ^1^HNMR (300 MHz, CDCl_3_): δ (ppm), 1.44 (3H, t, CH_3_), 2.68 (3H, s, CH_3_), 4.45 (2H, q, CH_2_), 4.63 (2H, s, CH_2_), 7.24–8.52 (10H, m, aromatic). MS (ESI) *m*/*z* [M + H]^+^ 365.1293, calcd. *m*/*z* 365.1318. FTIR (ATR, selected lines): ν (cm^−1^) 651 (C-S), 1686 (C=O).

*Ethyl 4-methyl-6-{[(2-methylphenyl)methyl]sulfanyl}-2-phenylpyrimidine-5-carboxylate* (**2b**) Product characterization: yield, 3.11 g, 82.28%; white solid; melting point, 105–106 °C; ^1^HNMR (300 MHz, DMSO-d_6_): δ (ppm), 1.30 (3H, t, CH3), 2.36 (3H, s, CH3), 2.57 (3H, s, CH3), 4.34 (2H, q, CH2), 4.62 (2H, s, CH2), 7.11–8.46 (9H, m, aromatic). MS (ESI) *m*/*z* [M + H]^+^ 379.1447, calcd. *m*/*z* 379.1474. FTIR (ATR, selected lines): ν (cm^−1^) 651 (C-S), 1698 (C=O).

#### 2.1.2. Preparation of Compounds **2c** and **2d**–**2j**

Ethyl 4-amino-6-methyl-2-phenylpyrimidine-5-carboxylate (**2c**) 

Ethyl 4-hydroxy-6-methyl-2-phenylpyrimidine-5-carboxylate (**1b**) (2.58 g, 10 mmol) [16] was placed in a round bottom flask, and 10 mL of POCl_3_ was added. The mixture was refluxed for 3 h, poured slowly into 100 mL of icy water, and extracted three times with 25 mL of CHCl_3_. The extracts were combined, dried with anhydrous MgSO_4_ for 30 min, and concentrated with a rotary evaporator. Then the solution of the crude product in 10 mL of THF was saturated with ammonia for 3 h, then poured into 100 mL of 2% HCl and extracted three times with 25 mL of CHCl_3_. The extracts were combined and dried with 5 g of anhydrous MgSO_4_ for 30 min. The drying agent was filtered off, and the solvent was removed with a rotary evaporator. The crude product was purified by column chromatography on silica gel using CHCl_3_ as eluent and dried at atmospheric pressure. The purity of the product was verified by TLC using chloroform as eluent.

Product characterization: yield, 2.30 g, 89.15%; beige solid; melting point, 127 °C; ^1^HNMR (300 MHz, CDCl_3_): δ (ppm), 1.43 (3H, t, CH_3_), 2.80 (3H, s, CH_3_), 4.41 (2H, q, CH_2_), 7.46–8.45 (5H, m, aromatic). MS (ESI) *m*/*z* [M + H]^+^ 258.1253, calcd. *m*/*z* 258.1237. FTIR (ATR, selected lines): ν (cm^−1^) 1674 (C=O), 3263 (NH_2_), 3187 (NH_2_).

Other methods for obtaining the compound were also reported [17].

#### 2.1.3. General Procedure for the Preparation of **2d**–**2j**

Ethyl 4-hydroxy-6-methyl-2-phenylpyrimidine-5-carboxylate (**1b**) (2.58 g, 10 mmol) [16] was placed in a round bottom flask, and 10 mL of POCl_3_ was added. The mixture was refluxed for 3 h, poured slowly into 100 mL of icy water, and extracted three times with 25 mL of CHCl_3_. The extracts were combined, dried with anhydrous MgSO_4_ for 30 min, and concentrated with a rotary evaporator. Then the crude product was dissolved in a mixture of 10 mL of methanol and 5 mL of triethylamine, and an appropriate substituted primary alkyl amine (12 mmol) was added. The reaction mixture was stirred at room temperature for 24 h. After this time, the precipitate was filtered off, washed with 5 mL of cold methanol, and dried at atmospheric pressure. The crude product was purified by column chromatography on silica gel using chloroform as eluent. The purity of the product was verified by TLC using chloroform as eluent.

*Ethyl 4-(ethylamino)-6-methyl-2-phenylpyrimidine-5-carboxylate* (**2d**). Product characterization: yield, 2.08 g, 72.73%; white solid; melting point, 54–55 °C; ^1^HNMR (300 MHz, CDCl_3_): δ (ppm), 1.34 (3H, t, CH_3_), 1.44 (3H, t, CH_3_), 2.75 (3H, s, CH_3_), 3.66–3.76 (2H, m, CH_2_), 4.40 (2H, q, CH_2_), 7.46–8.51 (5H, m, aromatic). MS (ESI) *m*/*z* [M + H]^+^ 286.1598, calcd. *m*/*z* 286.1550. FTIR (ATR, selected lines): ν (cm^−1^) 1670 (C=O), 3327 (NH).

*Ethyl 4-methyl-2-phenyl-6-(propylamino)pyrimidine-5-carboxylate* (**2e**). Product characterization: yield, 2.06 g, 68.67%; white solid; melting point, 45 °C; ^1^HNMR (300 MHz, CDCl_3_): δ (ppm), 1.03 (3H, t, CH_3_), 1.42 (3H, t, CH_3_), 1.72 (2H, sx, CH_2_), 2.75 (3H, s, CH_3_), 3.63 (2H, q, CH_2_), 4.39 (2H, q, CH_2_), 7.46–8.49 (5H, m, aromatic). MS (ESI) *m*/*z* [M + H]^+^ 300.1726, calcd. *m*/*z* 300.1707. FTIR (ATR, selected lines): ν (cm^−1^) 1674 (C=O), 3327 (NH).

*Ethyl 4-methyl-2-phenyl-6-[(prop-2-en-1-yl)amino]pyrimidine-5-carboxylate* (**2f**). Product characterization: yield, 1.99 g, 66.68%; white solid; melting point, 82–83 °C; ^1^HNMR (300 MHz, CDCl_3_): δ (ppm), 1.44 (3H, t, CH_3_), 3.21 (3H, s, CH_3_), 4.38 (2H, t, CH_2_), 4.46 (2H, q, CH_2_), 5.30 (2H, t, CH_2_), 5.92–6.05 (1H, m, CH), 7.53–8.67 (5H, m, aromatic). MS (ESI) *m*/*z* [M + H]^+^ 298.1530, calcd. *m*/*z* 298.1550. FTIR (ATR, selected lines): ν (cm^−1^) 1666 (C=O), 3299 (NH).

*Ethyl 4-methyl-2-phenyl-6-[(propan-2-yl)amino]pyrimidine-5-carboxylate* (**2g**). Product characterization: yield, 2.17 g, 72.33%; white solid; melting point, 108–109 °C; ^1^HNMR (300 MHz, CDCl_3_): δ (ppm), 1.34 (6H, d, 2xCH_3_), 1.44 (3H, t, CH_3_), 2.74 (3H, s, CH_3_), 4.39 (2H, t, CH_2_), 4.58 (1H, sx, CH), 7.47–8.49 (5H, m, aromatic), 8.21 (1H, broad, NH). MS (ESI) *m*/*z* [M + H]^+^ 300.1711, calcd. *m*/*z* 300.1707. FTIR (ATR, selected lines): ν (cm^−1^) 1674 (C=O), 3299 (NH).

*Ethyl 4-(tert-butylamino)-6-methyl-2-phenylpyrimidine-5-carboxylate* (**2h**). Product characterization: yield, 2.43 g, 77.39%; white solid; melting point, 103 °C; ^1^HNMR (300 MHz, CDCl_3_): δ (ppm), 1.42 (3H, t, CH_3_), 1.58 (9H, s, 3xCH_3_), 2.71 (3H, s, CH_3_), 4.37 (2H, q, CH_2_), 7.46–8.47 (5H, m, aromatic). MS (ESI) *m*/*z* [M + H]^+^ 314.1836, calcd. *m*/*z* 314.1863. FTIR (ATR, selected lines): ν (cm^−1^) 1666 (C=O), 3287 (NH).

*Ethyl 4-[(2-hydroxyethyl)amino]-6-methyl-2-phenylpyrimidine-5-carboxylate* (**2i**). Product characterization: yield, 2.25 g, 74.50%; white solid; melting point, 76–77 °C; ^1^HNMR (300 MHz, CDCl_3_): δ (ppm), 1.44 (3H, t, CH_3_), 2.77 (3H, s, CH_3_), 3.82–3.93 (4H, m, CH_2_-CH_2_) 4.40 (2H, q, CH_2_), 7.47–8.44 (5H, m, aromatic), 8.73 (1H, s, broad, NH). MS (ESI) *m*/*z* [M + H]^+^ 302.1482, calcd. *m*/*z* 302.1499. FTIR (ATR, selected lines): ν (cm^−1^) 1678 (C=O), 3319 (NH).

*Ethyl 4-(cyclohexylamino)-6-methyl-2-phenylpyrimidine-5-carboxylate* (**2j**). Product characterization: yield, 2.01 g, 59.12%; white solid; melting point, 72–73 °C; ^1^HNMR (300 MHz, CDCl_3_): δ (ppm), 1.25–2.15 (10H, m, 5xCH_2_, cykloheksyl; 3H, t, CH_3_), 2.77 (3H, t, CH_3_), 4.30 (1H, m, CH), 4.40 (2H, q, CH_2_), 7.46–7.52 (3H, m, aromatic), 8.40 (1H, broad, NH), 8.45–8.49 (2H, m, aromatic). MS (ESI) *m*/*z* [M + H]^+^ 340.2053, calcd. *m*/*z* 340.2020. FTIR (ATR, selected lines): ν (cm^−1^) 1670 (C=O), 3307 (NH).

#### 2.1.4. General Procedure for the Preparation of **3a**–**3h**

First, 2 mmol of proper ester 2 was dissolved in 20 mL of THF. Then, the mixture was cooled to 0 °C, and LiAlH_4_ (0.19 g, 5 mmol) was added gradually in small quantities. After 1 h, 25 mL of CHCl_3_ was added; then the mixture was poured into the icy water (100 mL) and extracted three times with CHCl_3_ (50 mL). The extracts were combined and dried with 2 g of anhydrous MgSO_4_ for 30 min. The drying agent was filtered off, and the solvent was removed with a rotary evaporator. The crude product was crystallized from methanol. The purity of the product was monitored by TLC using a mixture of chloroform and ethyl ether (3:1; *v*/*v*) as eluent.

*[4-(benzylsulfanyl)-6-methyl-2-phenylpyrimidin-5-yl]methanol* (**3a**). Product characterization: yield, 0.45 g, 70.04%; white solid; melting point, 144–145 °C; ^1^HNMR (300 MHz, CDCl_3_): δ (ppm), 2.76 (3H, s, CH_3_), 4.64 (2H, s, CH_2_), 4.79 (2H, s, CH_2_), 7.28–8.54 (10H, m, aromatic). MS (ESI) *m*/*z* [M + H]^+^ 323.1189, calcd. *m*/*z* 323.1213. FTIR (ATR, selected lines): ν (cm^−1^) 639 (C-S), 3374 (OH).

*(4-methyl-6-{[(2-methylphenyl)methyl]sulfanyl}-2-phenylpyrimidin-5-yl)methanol* (**3b**). Product characterization: yield, 0.49 g, 72.44%; white solid; melting point, 147 °C; ^1^HNMR (300 MHz, CDCl_3_): δ (ppm), 2.44 (3H, s, CH_3_), 2.65 (3H, s, CH_3_), 4.67 (2H, s, CH_2_), 4.77 (2H, s, CH_2_), 7.10–8.52 (9H, m, aromatic). MS (ESI) *m*/*z* [M + H]^+^ 337.1343, calcd. *m*/*z* 337.1369. FTIR (ATR, selected lines): ν (cm^−1^) 635 (C-S), 3343 (OH).

*(4-amino-6-methyl-2-phenylpyrimidin-5-yl)methanol* (**3c**). Product characterization: yield, 0.31 g, 72.09%; white solid; melting point, 177–178 °C; ^1^HNMR (300 MHz, DMSO-d_6_): δ (ppm), 2.40 (3H, s, CH_3_), 4.47 (2H, d, CH_2_), 4.93 (1H, t, OH), 6.58 (2H, s, broad, NH_2_), 7.41–8.30 (5H, m, aromatic). MS (ESI) *m*/*z* [M + H]^+^ 216.1110, calcd. *m*/*z* 216.1131. FTIR (ATR, selected lines): ν (cm^−1^) 3183 (OH), 3311 (NH_2_), 3418 (NH_2_).

*[4-(ethylamino)-6-methyl-2-phenylpyrimidin-5-yl]methanol* (**3d**). Product characterization: yield, 0.42 g, 85.71%; white solid; melting point, 191 °C; ^1^HNMR (300 MHz, DMSO-d_6_): δ (ppm), 1.22 (3H, t, CH_3_), 2.39 (3H, s, CH_3_), 3.54 (2H, q, CH_2_), 4.49 (2H, d, CH_2_), 5.00 (1H, t, OH), 6.77 (1H, broad, NH), 7.43–8.36 (5H, m, aromatic). MS (ESI) *m*/*z* [M + H]^+^ 244.1458, calcd. *m*/*z* 244.1444. FTIR (ATR, selected lines): ν (cm^−1^) 3239 (OH, broad), 3366 (NH).

*[4-methyl-2-phenyl-6-(propylamino)pyrimidin-5-yl]methanol* (**3e**). Product characterization: yield, 0.39 g, 76.47%; white solid; melting point, 158–159 °C; ^1^HNMR (300 MHz, DMSO-d_6_): δ (ppm), 0.94 (3H, t, CH_3_), 1.64 (2H, sx, CH_2_), 2.38 (3H, 3.47 (2H, q, CH_2_), 4.50 (2H, d, CH_2_), 5.03 (1H, t, OH), 6.77 (1H, broad, NH), 7.43–8.35 (5H, m, aromatic). MS (ESI) *m*/*z* [M + H]^+^ 258.1566 calcd. *m*/*z* 258.1601. FTIR (ATR, selected lines): ν (cm^−1^) 3263 (OH), 3414 (NH).

*{4-methyl-2-phenyl-6-[(prop-2-en-1-yl)amino]pyrimidin-5-yl}methanol* (**3f**). Product characterization: yield, 0.37 g, 72.55%; white solid; melting point, 174–175 °C; ^1^HNMR (300 MHz, DMSO-d_6_): δ (ppm), 2.40 (3H, s, CH_3_), 4.16 (2H, t, CH_2_), 4.52 (2H, d, CH_2_), 5.03-5.27 (3H, m, CH_2_; OH), 5.94–6.07 (1H, m, CH), 6.94 (1H, t, NH), 7.43–8.33 (5H, m, aromatic). MS (ESI) *m*/*z* [M + H]^+^ 256.1402, calcd. *m*/*z* 256.1444. FTIR (ATR, selected lines): ν (cm^−1^) 3259 (OH, broad), 3347 (NH).

*{4-methyl-2-phenyl-6-[(propan-2-yl)amino]pyrimidin-5-yl}methanol* (**3g**). Product characterization: yield, 0.40 g, 78.43%; white solid; melting point, 160 °C; ^1^HNMR (300 MHz, DMSO-d_6_): δ (ppm), 1.25 (6H, d, 2xCH_3_), 2.38 (3H, s, CH_3_), 4.42 (1H, sx, CH), 4.50 (2H, d, CH_2_), 5.09 (1H, t, OH), 6.42 (1H, d, NH), 7.43–8.34 (5H, m, aromatic). MS (ESI) *m*/*z* [M + H]^+^ 258.1605, calcd. *m*/*z* 258.1601. FTIR (ATR, selected lines): ν (cm^−1^) 3215 (OH), 3386 (NH).

*[4-(tert-butylamino)-6-methyl-2-phenylpyrimidin-5-yl]methanol* (**3h**). Product characterization: yield, 0.37 g, 68.52%; white solid; melting point, 173–174 °C; ^1^HNMR (300 MHz, DMSO-d_6_): δ (ppm), 1.52 (9H, s, 3xCH_3_), 2.36 (3H, s, CH_3_), 4.50 (2H, d, CH_2_), 5.31 (1H, t, OH), 6.31 (1H, s, NH), 7.45–8.33 (5H, m, aromatic). MS (ESI) *m*/*z* [M + H]^+^ 272.1722, calcd. *m*/*z* 272.1757. FTIR (ATR, selected lines): ν (cm^−1^) 3124 (OH), 3331 (NH).

*2-{[5-(hydroxymethyl)-6-methyl-2-phenylpyrimidin-4-yl]amino}ethan-1-ol (***3i***).* Product characterization: yield, 0.44 g, 83.72%; white solid; melting point, 141 °C; ^1^HNMR (300 MHz, DMSO-d_6_): δ (ppm), 2.40 (3H, s, CH_3_), 3.55–3.69 (4H, m, CH_2_-CH_2_), 4.50 (2H, d, CH_2_), 4.79 (1H, t, OH), 5.07 (1H, t, OH), 6.82 (1H, s, broad, NH), 7.44–8.36 (5H, m, aromatic). MS (ESI) *m*/*z* [M + H]^+^ 260.1392, calcd. *m*/*z* 260.1394. FTIR (ATR, selected lines): ν (cm^−1^) 3147 (OH, broad).

*[4-(cyclohexylamino)-6-methyl-2-phenylpyrimidin-5-yl]methanol* (**3j**). Product characterization: yield, 0.49 g, 83.05%; white solid; melting point, 143 °C; ^1^HNMR (300 MHz, DMSO-d_6_): δ (ppm), 1.17–2.04 (10H, m, 5xCH_2_), 2.38 (3H, s, CH_3_), 4.10 (1H, s, CH), 4.52 (2H, d, CH_2_), 5.17 (1H, t, OH), 6.47 (1H, d, NH), 7.40–8.40 (5H, m, aromatic). MS (ESI) *m*/*z* [M + H]^+^ 298.1924, calcd. *m*/*z* 298.1914. FTIR (ATR, selected lines): ν (cm^−1^) 3243 (OH), 3362 (NH).

### 2.2. X-ray Structural Studies

Compounds **3c** and **3e**–**3h** were dissolved in hot methanol or acetone, and their crystals were obtained by slow evaporation of the solvent at room temperature. Crystals suitable for X-ray diffraction analysis appeared after 1–5 days. Single-crystal X-ray diffraction data were collected with Mo-Kα (λ = 0.71073 Å) (for **3c**, **3e**, and **3g**–**3h**) or Cu-Kα (λ = 1.5418 Å) (for **3f**) radiations and ω-scan modes using automated four-circle diffractometers with CCD detectors: Kuma KM4 (for **3c** and **3h**), Xcalibur R (for **3e** and **3g**), and Xcalibur PX (for **3f**). The data were measured at 200(2) K (for **3c**) or 100(2) K (for **3e**–**3h**) using an Oxford Cryosystems open-flow nitrogen cryostat. This paper provides the selected crystallographic data and structure refinements for compounds **3c** and **3e**–**3h** in Appendix A. The CrysAlisPro software package [18] was used for data collection, cell refinement, data reduction, and analysis. The data of **3f** were corrected for absorption by the analytical method. The crystal structures were solved by direct methods using SHELXS-97 [19] and refined by a full-matrix least-squares technique on F^2^ with SHELXL-2013 (and further also with SHELXL-2018) [20]. All non-H atoms were refined with anisotropic displacement parameters. During the refinement for **3h**, one of two crystallographically independent molecules exhibited disorder over two positions related by a pseudo mirror plane (denoted as B and C) and was refined in the same site occupation factors of 0.5. SAME instruction was used in the refinement procedure equal to the restrained geometrical parameters of the disordered molecule of **3h** (equivalent bond distances and angles). Additionally, two C41C and C43C atoms were constrained with EADP instruction and refined with the same fractional coordinates. All H atoms were initially found in the difference Fourier maps and refined isotropically. In the final refinement cycles, they were included from the geometry of the molecules and refined isotropically using a riding model, with C–H = 0.95–0.99 Å and U_iso_(H) = 1.2U_eq_(C) for CH and CH_2_ or U_iso_(H) = 1.5U_eq_(C) for CH_3_, N–H = 0.88 Å and O–H = 0.84 Å, and U_iso_(H) = 1.2U_eq_(N) and 1.5U_eq_(O), respectively. All figures were prepared using the DIAMOND program [21]. Analysis of the intra- and intermolecular interactions was performed using the PLATON program [22].

The supplementary crystallographic data for this article (CCDC 2102842-2102846) were deposited in the Cambridge Crystallographic Data Centre at www.ccdc.cam.ac.uk/data_request/cif (accessed on 8 November 2021) and can be obtained free of charge, by e-mailing data_request@ccdc.cam.ac.uk, or by contacting the Cambridge Crystallographic Data Centre, 12 Union Road, Cambridge CB2 1EZ, UK (fax: +44(0)1223-336033).

### 2.3. Biological Activity Assays

#### 2.3.1. Chemicals

All reagents for cell culture, unless otherwise stated, were purchased from Merck (Darmstadt, Germany). Stock solutions were made 100x concentrated in pure DMSO (dimethyl sulfoxide) (Chempur, Poland) and freshly diluted in appropriate complete cell culture medium on the experiment day. Cell cultures were acquired from the European Collection of Authenticated Cell Cultures (ECACC). Reagents for microbiology assays were purchased from Oxoid (Basingstoke, UK).

#### 2.3.2. Cell Culture

For cytotoxicity assays, the following cell lines were used: human renal proximal tubule epithelial cells (RPTEC-MTOX1030), epithelioid cervix carcinoma (HeLa-93021013), colon adenocarcinoma (Caco-2-86010202), hepatocarcinoma (HepaRG-MTOX1010), gastric adenocarcinoma (AGS-89090402), and glioblastoma (A172-88062428) cells were cultured in appropriate media according to the European Collection of Authenticated Cell Cultures (ECACC) recommendation. To all media (except for RPTEC), 2.5 µg mL^−1^ amphotericin B, 100 U mL^−1^ penicillin, and 100 µg mL^−1^ streptomycin were added. In addition, the RPTEC cell line was cultured in media with 30 μg mL^−1^ gentamicin and 15 ng mL^−1^ amphotericin B. Cell cultures were incubated at 37 °C in a humidified atmosphere with 5% CO_2_ for at least three passages until performing experiments. All cell lines were grown in monolayers; thus, detachment with 0.25% trypsin solution was required for further assays.

#### 2.3.3. Neutral Red Uptake Assay

The cells were put in 96-well plates (1 × 105 cells per mL) of appropriate culture medium (100 µL per well). Before a general experiment, cells were attached for 24 h and then treated with various concentrations (500, 250, 100, and 10 µM) of pyrimidine derivatives and subsequently grown in 200 µL medium volume. Controls with 1% DMSO, culture medium, and 1 µM staurosporine were running simultaneously with the pyrimidine-derivatives-treated cultures. The cells were incubated for 24, 48, and 72 h at 37 °C in a humidified 5% CO_2_/95% air incubator. The neutral red uptake assay was performed according to Repetto et al. [23]. This method uses the ability of the living cells to accumulate dye in their lysosomes. Briefly, at the end of each incubation time the cells were treated with neutral red solution (40 µg mL^−1^) for 2 h at 37 °C. Afterwards, the cells were washed with PBS, and the incorporated dye was solubilized with the extraction solution (50% ethanol, 49% H_2_O, 1% glacial acetic acid). The plates were gently shaken for 30 min at 37 °C. Finally, the absorbance was measured at 540 nm using a Thermo Scientific™ Multiskan™ GO Microplate Spectrophotometer (Thermo Fisher).

#### 2.3.4. Microbiology

Antimicrobial activity assays were performed on seven reference strains obtained from ATCC collection (Acinetobacter baumannii, Candida albicans, Enterococcus faecalis, Escherichia coli, Klebsiella pneumoniae, Pseudomonas aeruginosa, methicillin-resistant Staphylococcus aureus (MRSA)). Microorganisms were propagated on Tryptone Soy Agar (TSA) plates. Bacteria were incubated at 37 °C, and Candida albicans was incubated at 25 °C. After 24 h incubation, microorganisms were diluted in Tryptone Soy Broth (TSB) (bacteria to 0.005MF and C. albicans to 0.025MF) and seeded on 96-well plates, which already contained tested compounds at final concentrations from 256 µg mL^−1^ to 0.5 µg mL^−1^. Every plate included controls: TSB with strain, TSB with strain and solvent (1% DMSO), and TSB with strain and appropriate antibiotic. Microplates were incubated on a horizontal shaker at 500 rpm in 37 ± 1 °C for 24 h, and absorbance was measured by a microplate reader at 580 nm using the Thermo Scientific™ Multiskan™ GO Microplate Spectrophotometer (Thermo Fisher). Subsequently, 50 µL aliquots of 5 µL of 1% aqueous solution of 2,3,5-triphenyltetrazolium chloride (TTC) were added to each well to check the viability of microorganisms. Plates were inspected visually for color change after 24 h incubation.

#### 2.3.5. Statistical Analysis

The viability tests were conducted in quadruple wells (*n* = 4) for each condition. Values are presented as mean ± standard deviation (SD). Significant differences were calculated using one-way ANOVA, followed by post hoc comparison, and the half-maximal inhibitory concentration (IC50) was obtained by nonlinear regression using GraphPad Prism 9.1 for Windows, GraphPad Software, La Jolla, California, USA, www.graphpad.com. A *p*-value < 0.05 was considered statistically significant. All results were compared with the control with 1% DMSO, which was considered to be 100%.

### 2.4. In Silico Analysis

For the analysis, compounds **3a**, **3h**, and **3g** were chosen due to their half-maximal inhibitory concentration (IC50) measured after 72 h incubation with cancerous cell lines.

#### 2.4.1. ADME Analysis

ADME analysis of compounds **3a**, **3g**, and **3h** was performed via SwissADME [24], a freely available software provided by the Swiss Institute of Bioinformatics. The software provides information about estimated predictors, such as basic physiochemical properties, lipophilicity, water solubility, pharmacokinetics, druglikeness, and medicinal chemistry (Table 4). Physiochemical properties are relevant due to crossing biological barriers. Lipophilicity is very important for pharmacokinetics drug discovery. Orally active drugs should consist of no more than 5 hydrogen bond donors less than 10 hydrogen bond acceptors, molecular weight of less than 500 daltons, and an n-octanol and water partition coefficient lesser than 5 [25]. Water solubility is connected to the oral admission of the drug. Water solubility is the major property of medicine absorption. Being a substrate of the permeability glycoprotein is a predictor of pharmacokinetics. Knowledge about the interaction of the compound and cytochromes P450 gives information about effective drug elimination through metabolic biotransformation. The last predictor-medicinal chemistry-supports structural drug discovery. It informs about problematic structural fragments inside the investigated compound.

#### 2.4.2. Molecular Docking Analysis

Estimation of probable macromolecular targets was performed using SwissTargetPrediction [26], choosing *Homo sapiens* as a target species. SwissTargetPrediction is a freely available web tool provided by the Swiss Bioinformatics Institute.

The protein target structures listed above were downloaded from the Protein Data Bank (PDB). The inhibitors from the complexes were removed, as well as water molecules. The pH value for the protonation of both the ligand (compounds **3a**, **3g**, and **3h**) and the protein was set to 7.4 using PDB2PQR [27]. The input files were prepared using AutoDock tools [28]. The search space of each protein was defined from “center on hetero”. A box was placed on the geometric center of an existing ligand. The molecular docking was performed using AutoDock Vina [29]. The docking was performed for each target from the Table 5. presented in Section 3.4.

## 3. Results

### 3.1. Chemistry

The substrates **1a** and **1b** were prepared as described previously [15,16]. The benzylsulfanyl esters **2a** and **2b** were obtained using the method exploited in our previous publication (Figure 2) [14]. In short, substrate **1a** was dissolved in a methanolic solution of NaOH and coupled with appropriate benzyl chloride. The precipitate of ester was washed and dried at atmospheric pressure.

Aminoesters were obtained by the reaction of the product of chlorination of the substrate 1b with gaseous ammonia in THF (**2c**) or proper alkyl amines in the mixture of trimethylamine and methanol (**2d**–**2j**) (Figure 3) and purified via column chromatography using CHCl_3_ as eluent.

Final products **3a**–**3j** were obtained by reduction of esters **2a**–**2j** with LiAlH_4_ in THF (Figure 4) and purified by column chromatography (**3a** and **3b**) or crystallization (**3c**–**3j**).

Using the presented set of reactions, we obtained a series of various 5-hydroxymethylpyrimidines that vary at the 4-position. Structural variations of the compounds were applied to draw conclusions about the structure–activity relationship.

### 3.2. Crystal Structures of Compounds ***3c*** and ***3e***–***3h***

Compounds **3c** and **3e**–**3g** crystallize in a common *P*2_1_/*c* space group of the monoclinic system with Z = 4 and one crystallographically independent molecule in the asymmetric unit (Figure 5a–d). In contrast to them, it was found that compound **3h** crystallizes in the noncentrosymmetric orthorhombic *Pca*2_1_ space group, with Z = 8, and comprises two crystallographically independent molecules in the asymmetric unit, one ordered molecule (denoted as A) (Figure 5e) and a second disordered molecule over two positions being related by a pseudo mirror plane (hereafter referred to as B and C). Selected crystallographic data and structure refinements are presented in Appendix A, and comparison of selected geometrical parameters for compounds **3c** and **3e**–**3h** are tabulated in Appendix A of this paper.

All molecules of compounds **3c** and **3e**–**3h** contain the same 5-hydroxymethyl-2-phenylpyrimidine core and vary at the 4-position, in particular containing amino (**3c**), propylamino (**3e**), allylamino (**3f**), isopropylamino (**3g**), and *tert*-butylamino (**3g**) groups. All molecules have a conformation with the planar pyrimidine ring (with rms. deviation range of 0.008–0.020 Å) and the substituent in the 2-position twisted relative to it by 26.4 (2)° in **3c**, 9.5 (2)° in **3e**, 10.9 (2)° in **3f**, 15.7 (2)° in **3g**, and 18.5–20.6 (2)° in **3h**, respectively. In cases of secondary amines, the heterocyclic N3 atom and the C41 atom of the amino group are nearly located on the same side of the molecule, as shown by the N3–C4–N4–C41 torsion angle (−6.8 (2)°, 17.1 (2)°, 0.4 (2)°, and −12 (3)° in **3e**, **3f**, **3g**, and **3h** (see Appendix A of this paper). The hydroxyl moiety [O(H)-C = 1.418–1.441 Å] is in a *gauche* conformation with respect to the pyrimidine C4 atom, which is reflected by the values of the O1–C51–C5–C4 torsion angle of −53.9 (2)°, 63.3 (2)°, 64.5 (2)°, −62.9 (2)°, and 49.0 (6)° (or 50 (2)° and 49 (2)°) for **3c**, **3e**, **3f**, **3g**, and the molecule A (or in the positions B and C of the second molecule) of **3h**, respectively. Except for **3f**, the orientation of the amino and hydroxyl groups to each other is stabilized by the intramolecular N4−H···O1 hydrogen bond (see Table 1), which closes a six-membered ring with S(6) motif [30].

In the crystal structures, except for **3f**, each molecule is connected via O–H···N hydrogen bonds (Table 1), between the hydroxyl O1 (as donors) and the pyrimidine N1 atoms (as acceptors), to form a C(6) chain along the c-axis. Representative one-dimensional chains formed through intermolecular O–H···N interactions in the crystal structures of **3c** and **3g** are presented in Figure 6. In case of **3f**, such O–H···N hydrogen bond interactions lead to the formation of a dimer. These dimers interact further via intermolecular N–H···O hydrogen bonds, between the amine N4 (as donors) and the hydroxyl O1 atoms (as acceptors), resulting in a ribbon of molecules running along the c-axis direction (Figure 7). In **3c**, a 2D hydrogen-bonded network is created by the connection of neighboring chains through N−H∙∙∙N hydrogen-bonding interactions, involving the amine N4 (as donors) and the pyrimidine N3 atoms (as acceptors) (see Appendix A). In all cases, the crystal structures are stabilized by weak C–H∙∙∙π and also aromatic π–π stacking interactions. In turn, for **3g**, additional stabilization is also provided by weak C–H···O hydrogen bonding interactions, involving the isopropyl (as donors) and hydroxyl (as acceptors) groups (Table 1).

### 3.3. Biological Activity Analysis

#### 3.3.1. Cytotoxic and Antiproliferative Effect

Ten pyrimidine derivatives, **3a**–**3j**, were tested for the cytotoxic properties on the RPTEC cell line with the neutral red uptake assay. Results are shown in Figure 8. Compounds **3b** (500 µM = 34%, 250 µM = 46%, 100 µM = 44%, 10 µM = 91%) and **3j** (500 µM = 16%, 250 µM = 70%, 100 µM = 28%, 10 µM = 89%) decreased the cell viability at lower concentrations (100 and 10 µM) and, therefore, were excluded from further analysis on cancerous cell lines and microorganisms.

In further research, pyrimidine derivatives were tested for their anticancer activity. Compounds in a wide concentration range (10–250 µM) were applied on cells. Viability and proliferative abilities were tested with the neutral red uptake assay and assessed by comparing with the control with solvent (1% DMSO). Calculated IC50 values for all tested compounds are presented in Table 2. The highest cytotoxic properties were found after HepaRG treatment with **3h** (132.3 µM). This cell line was found to be the most susceptible for tested compounds, as three from eight pyrimidine derivatives decreased cell viability to the sufficient level to calculate IC50.

#### 3.3.2. Antimicrobial Activity

The antimicrobial activities of tested pyrimidine derivatives are presented in Table 3 as minimal inhibitory concentration (MIC). MICs were determined for *A. baumannii* (**3e**, **3g**, and **3h** = 256 µg/mL), *C. albicans* (**3a** = 256 µg/mL), *E. coli* (**3g** = 128 µg/mL), *E. faecalis* (**3e** = 256 µg/mL), and MRSA (**3f** = 256 µg/mL). Treatment with TTC indicated no MBC/MFC activity.

### 3.4. In Silico Studies

#### 3.4.1. ADME Analysis

For the in silico analysis, compounds **3a**, **3h**, and **3g** were chosen due to their preferable IC50 measured after 72 h incubation with cancer cell lines. ADME analysis was performed regarding six predictors pointed in 2.4.1 [24]. In Table 4, specific details for the chosen compound are presented. 

#### 3.4.2. Molecular Docking Analysis

The binding pose and affinity between a ligand and an enzyme are very important pieces of information for computer-aided drug design. In the initial stage of a drug discovery project, this information is often obtained by using molecular docking methods. To conduct an effective docking procedure, some conditions have to be fulfilled. The structure of a target protein extracted from X-ray crystallography gives the possibility to omit protein structural changes during the process of binding the same ligand. According to Jones et al. [50], the resolution of the protein–ligand complex should be below 2.5 Å. Poor resolution structures result in more incorrect conformations of the ligand generated. Moreover, the complex should have only one ligand in the active site. The results of the target prediction are shown in Table 5. The docking was performed for each target from Table 5. The results are presented for one target—the one for which the docking results were better.

The results of docking present a prediction of the ligand binding modes, classified by the scoring function as the best. The proteins are presented as “surface”, and each ligand is presented as lines (Figure 9, Figure 10 and Figure 11). 

## 4. Discussion

The starting esters were obtained by the reaction of 4-sulfanylpyrimidine (**1a**) with proper benzyl chloride in a methanolic solution of NaOH (**2a** and **2b**) or by coupling the product of chlorination of 1b with gaseous ammonia (**2c**) or aliphatic amines (**2d**–**2j**). Their reduction yielded a series of 10 new 5-hydroxymethylpyrimidines (**3a**–**3j**), which were the subject of further research. The structures of the obtained compounds were established by spectroscopy techniques, and several of them were identical with solid-state structures obtained during X-ray measurements.

The molecular structures of **3c** and **3e**–**3h** were studied by single-crystal X-ray diffraction. Compounds **3c** and **3e**–**3g** crystallize in the space group *P*2_1_/*c* with one molecule in the asymmetric unit, whereas compound **3h** crystallizes in the space group *Pca*2_1_ with two molecules in their asymmetric units. The preferred *gauche* orientation of the hydroxyl O1 and the pyrimidine C4 atoms was confirmed by the values of the O1–C51–C5–C4 torsion angle of 49.0–64.5 (2)°. Such conformation is stabilized by an intramolecular N–H···O interaction, which closes a six-membered ring with S(6) motif with the exception of 3f. It seems that the smaller the absolute value of the O1–C51–C5–C4 torsion angle is, the stronger the intramolecular N–H···O bond becomes. A similar synthon (S(6)), which is formed by an intramolecular interaction of another type (N–H···N), can be found in previously studied crystal structures of 2-phenylpyrimidine-4-amine derivatives [13,51]. Generally, in all structures, a similar hydrogen-bonded network can be observed, which involves mainly –OH moieties as donors (in intermolecular interactions) and acceptors (in intramolecular interactions, except for **3f**). Despite the similarity, it should also be mentioned that the structure of **3g** is subtly different. In the cases of **3g**, the amino group is nearly coplanar to the pyrimidine ring, as shown by the N3–C4–N4–C41 torsion angle 0.4 (2)°, and additional stabilization of the crystal packing is also provided by weak C–H···O hydrogen bonding interactions, involving the isopropyl and hydroxyl groups.

All the obtained 5-hydroxymethylpyrimidines were tested for their cytotoxicity against normal cells on the RPTEC cell line with a neutral red uptake assay. Generally, pyrimidines possessing a 4-benzylsulfanyl group (**3a** and **3b**) exhibit stronger toxicity than their amino analogues (**3c**–**3j**). Considering our previous observation [14], it appears that the substitution in the phenyl ring of the benzyl group, with both electron-withdrawing (4-Cl) and electron-donating groups (2-CH_3_, **3b**), enhances their toxic properties. Compound **3b** decreased the cell viability in low concentration and, for this reason, was excluded from further biological investigations.

Pyrimidines with primary or secondary amino groups at the 4-position (**3c**–**3j**) show relatively low toxicity in a wide range of concentrations with the exception of compound **3j**. It seems that the substitution of the primary amino group of compound **3c** with n-alkyls (ethyl-3d, n-propyl-3e), unsaturated alkyl (allyl-3f), branched alkyls (isopropyl-3g, *t*-butyl-3h), or hydroxyalkyl groups (2-hydroxyethyl-3i) exerts an insignificant impact on their cytotoxicity. The substantial growth of toxicity can be seen in the case of compound **3j**, which possesses a sizable cyclohexyl substituent at the amino group and, because of that, was excepted from further tests.

The anticancer investigation showed that only three out of eight tested compounds demonstrated antineoplastic properties. Derivative **3a** with a 4-benzylsulfanyl group exhibited low toxicity to HepaRG and AGS cell lines. Among derivatives with 4-amino groups, only two exhibited moderate toxicity, namely, those with bulky substituents as isopropyl (**3g**) and *t*-butyl (**3h**) groups. At the same time, **3h** was found to reach the lowest IC50 among tested compounds (132.2 µM). On the other hand, **3g** demonstrated the broadest range of toxicity against cancer cell lines (HeLa, HepaRG, A172). As mentioned above, HepaRG was the most susceptible line to tested compounds. It is also noteworthy that the substitution of the primary amino group of 3c with various alkyl groups does not affect their anticancer properties, and the enhancement is evident only in the case of branched substituents.

Selected compounds for ADME analysis predict similar physiochemical properties and have comparable molecular weights, rotatable bonds, number of hydrogen bond donors/acceptors and water solubility, druglikeness, and medicinal chemistry properties. Due to the lipophilicity of all the investigated compounds, a partition coefficient was less than 5, and **3g** has the lowest partition coefficient between n-octanol and water. Moreover **3a** has the best predicted pharmacokinetic properties.

The docking of compound **3a** to the phosphodiesterase 10A binding site (Figure 9) resulted in nine generated poses of the ligand with the top-scored generated pose no. 4 with an affinity of −8.9 kcal/mol and an RMSD of 1.453 Å. The compound forms hydrogen bonding with Tyr514 and Gln716 in the binding site of the enzyme. The docking of compound **3g** to the phosphodiesterase 4B binding site (Figure 10) resulted in nine generated poses of the ligand with the top-scored generated pose no. 2 with an affinity of −8.9 kcal/mol and an RMSD of 1.015 Å. The compound forms hydrogen bonding with Gln443 and Asn395 in the binding site of the enzyme. The docking of compound **3h** to the adenosine A1 receptor binding site (Figure 11) resulted in nine generated poses of the ligand with the top-scored generated pose no. 6 with an affinity of −8.4 kcal/mol and an RMSD of 1.417 Å. The compound forms hydrogen bonding with Glu169 and Asn253, Phe168 in the binding site of the enzyme.

The tested compounds exhibited weak antimicrobial properties. Only **3a** with a 4-benzylsulfanyl group demonstrated any antifungal activity (MIC against *C. albicans* = 256 µg/mL), but it had no antibacterial features. Among the derivatives with a 4-amino group, compounds with smaller than three carbon chains exhibited no antimicrobial properties (**3c** with a primary amino group, **3d** with ethyl, and **3i** with 2-hydroxyethyl groups). The extension of the carbon chain slightly enhances the antibacterial potency of the reminder. The lowest MIC was observed in the case of **3g** with an isopropyl substituent against *E. coli* (MIC = 128 µg/mL). *A. baumannii* was the most sensitive bacterium to the tested compounds.

## 5. Conclusions

Based on our previous research, we obtained a series of novel 5-hydroxymethylpyrimidines, **3a**–**3j**, which vary in the 4-position by the reduction of proper esters, **2a**–**2j**, with LiAlH_4_. The compounds were characterized using spectroscopic and some of them (**3c** and **3e**–**3h**) also X-ray crystallography methods.

Except for **3h**, which crystallized in the *Pca*2_1_ space group, the other studied compounds crystallized in the *P*2_1_/*c* space group. The preferred gauche orientation of the –OH group with respect to the pyrimidine C4 atom is stabilized by an intramolecular N–H···O hydrogen bond, with the exception of **3f**. On the whole, a hydrogen-bonded network structure involves primarily –OH moieties as donors (in intermolecular interactions) and acceptors (in intramolecular interactions, except for **3f**). Furthermore, in the case of the crystal packing of **3g**, weak C–H···O hydrogen bonding interactions involving the isopropyl and hydroxyl groups are also observed.

The cytotoxic test against normal cells on the RPTEC cell line showed that compounds with benzyl sulfanyl groups at the 4-position are generally more toxic than those with amino groups at the 4-position, except for **3j**, which possesses a sizable cyclohexyl group. Thus, 2 out of 10 compounds were excluded from further investigation, **3b** with a 2-methylbenzyl sulfanyl group at the 4-position and **3j**, mentioned above.

We examined the anticancer activity of the compounds on various cancer cell lines, including HeLa, HepaRG, Caco-2, AGS, and A172, and their antimicrobial properties on six bacterial strains and one strain of fungi. On the whole, the compounds showed weak biological activity. Compound **3g** exhibited the broadest range of toxicity against cancer cell lines (HeLa, HepaRG, A172), while **3h** reached the lowest IC50 among tested compounds (132.2 µM against HepaRG). 

Furthermore, in silico analysis reported that ligands that interact with selected enzymes that have been used as docking targets might have moderate therapeutic activity. For instance, the tested compounds have similar values of binding affinities. Nevertheless, the docking of **3g** in the enzyme binding site showed the least value of RMSD.

Regarding microbial activity, only compound **3a** with a benzyl sulfanyl group at the 4-position showed any antifungal properties (MIC = 256 µg mL^−1^). Compound **3g** demonstrated the lowest MIC among the tested compounds (MIC = 128 µg mL^−1^ against *E. coli*).

It is worth mentioning that 5-hydroxymethylpyrimidines that possess bulky amino constituents at the 4-position (3g-isopropyl and 3h-*t*-butyl) showed the best anticancer and antibacterial properties among tested derivatives, which can be a valuable foothold for further research.

## Figures and Tables

**Figure 1 materials-14-06916-f001:**
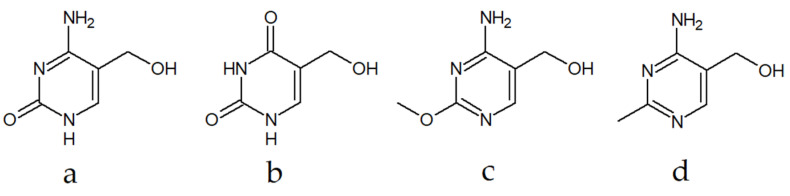
Naturally occurring 5-hydroxymethylpyrimidine derivatives: (**a**) 5-hydroxymethylcytosine; (**b**) 5-hydroxymethyluracil; (**c**) bacimethrin; (**d**) vitamin B_1_ (thiamine).

**Figure 2 materials-14-06916-f002:**
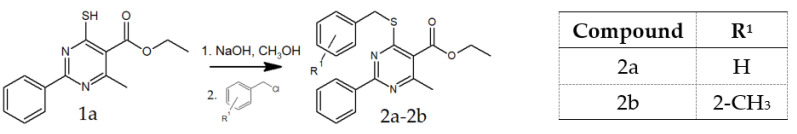
Synthetic routes of preparing compounds **2a** and **2b**.

**Figure 3 materials-14-06916-f003:**
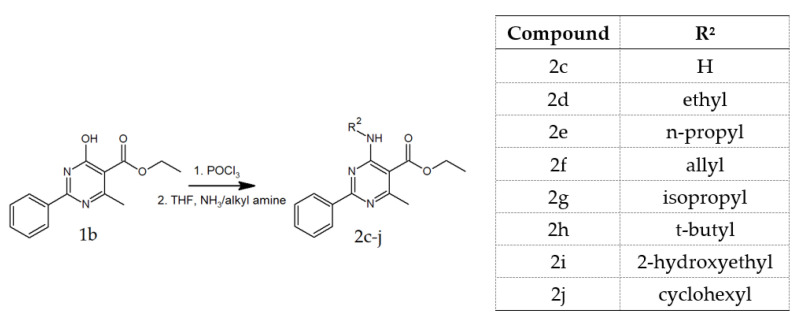
Synthetic routes of preparing compounds **2c**–**2j**.

**Figure 4 materials-14-06916-f004:**
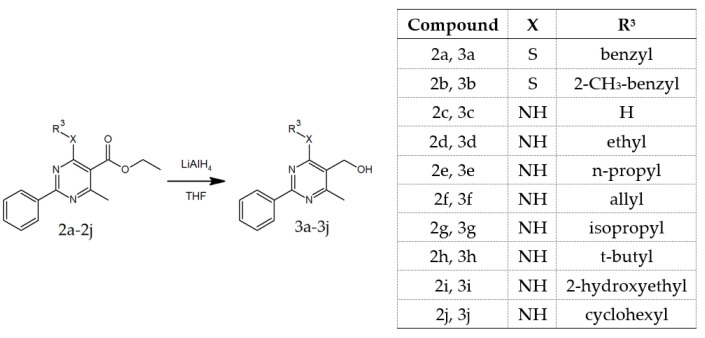
Synthetic route of preparing compounds **3a**–**3j**.

**Figure 5 materials-14-06916-f005:**
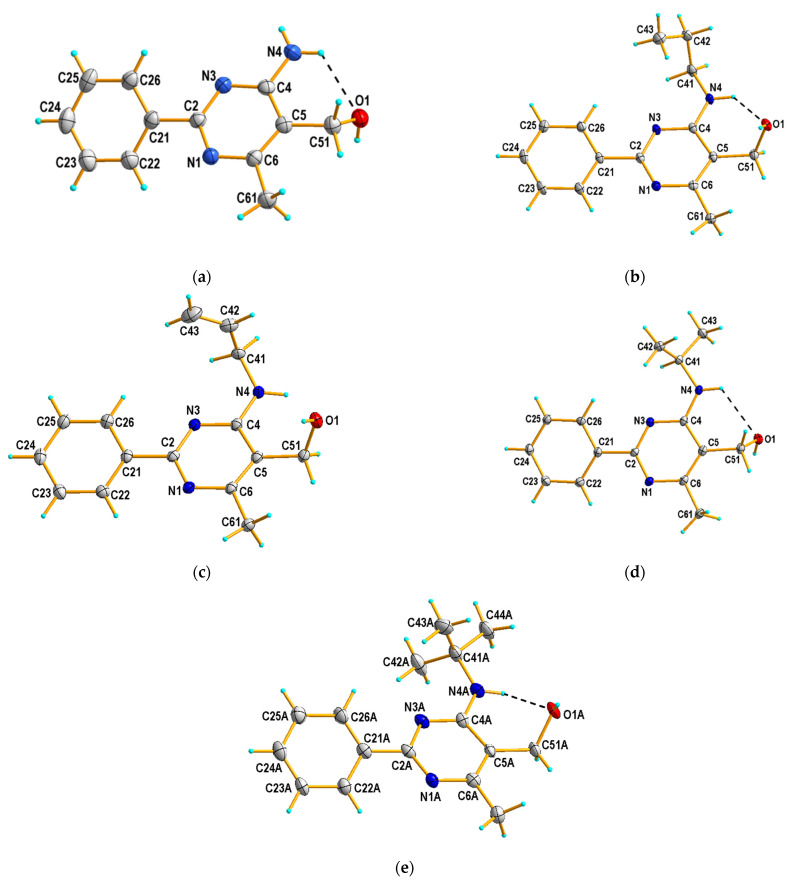
X-ray structures of **3c** (**a**), **3e** (**b**), **3f** (**c**), and **3g** (**d**) and one of two crystallographically independent molecules (denoted as A) of **3h** (**e**) (displacement ellipsoids at 50% probability level), with an atom-numbering scheme. Dashed lines (in black) show intramolecular N–H···O hydrogen bonds forming S(6) motifs.

**Figure 6 materials-14-06916-f006:**
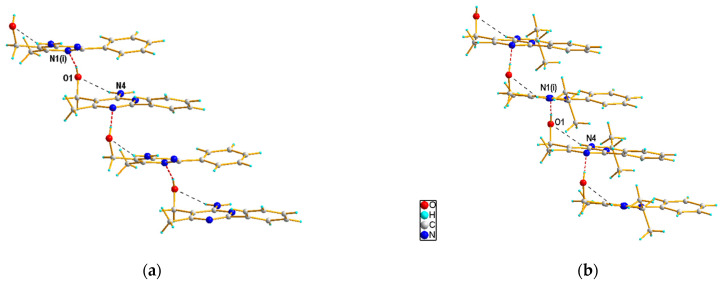
Part of the crystal structures of **3c** (**a**) and **3g** (**b**), showing the formation of the one-dimensional chain through O−H∙∙∙N (in red) intermolecular interactions. Dashed lines (in black) indicate intramolecular N−H∙∙∙O hydrogen bonds. The symmetry codes are as in Table 1.

**Figure 7 materials-14-06916-f007:**
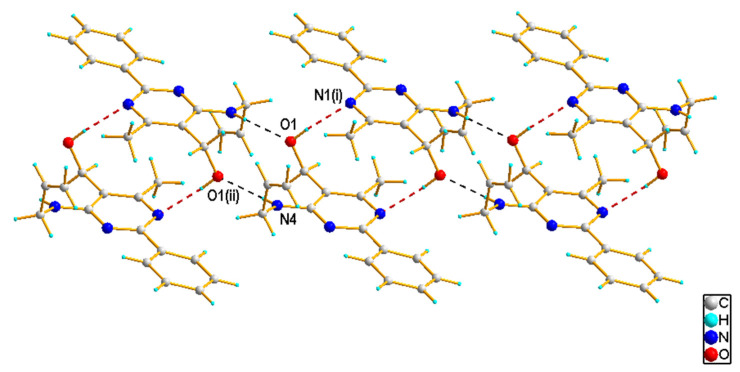
Part of the crystal structure of **3f**, showing the formation of the ribbon through intermolecular interactions, with O−H∙∙∙N in red and N–H···O in black. The symmetry codes are as in Table 1.

**Figure 8 materials-14-06916-f008:**
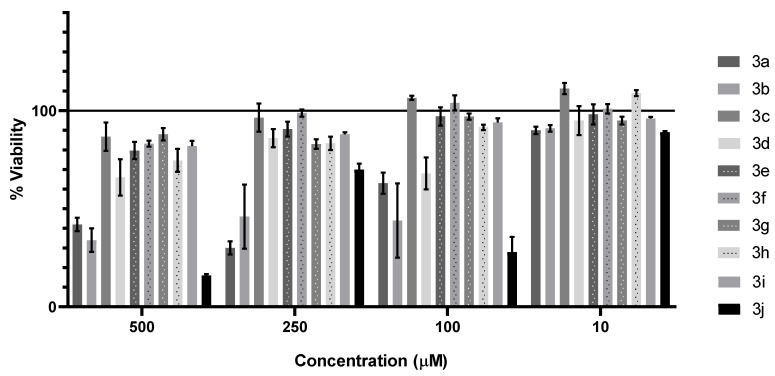
Viability of RPTEC cell line after 72 h treatment with tested compounds.

**Figure 9 materials-14-06916-f009:**
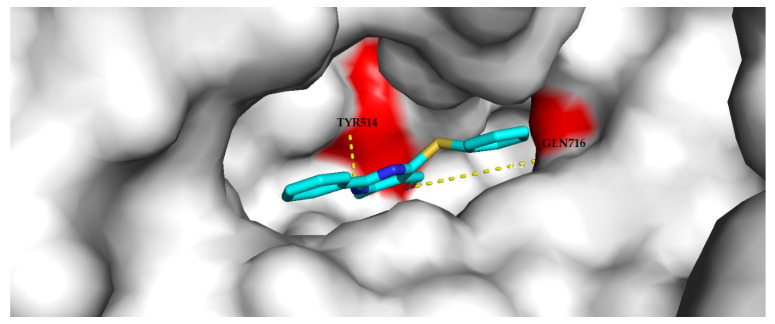
Compound **3a** in a phosphodiesterase 10A binding site. Yellow dashes indicate the hydrogen bonding with amino acids in the binding site (colored as red). The presented ligand was top-scored by the scoring function and obtained the lowest RMSD value (1.453 Å) after docking in AutoDock Vina.

**Figure 10 materials-14-06916-f010:**
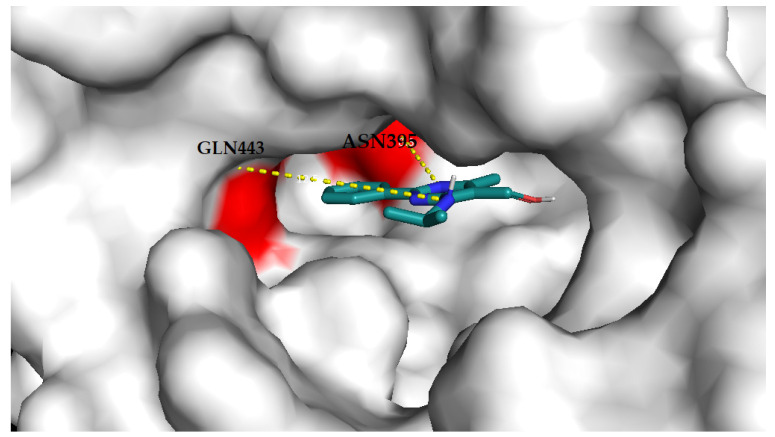
Compound **3g** in a phosphodiesterase 4B binding site. Yellow dashes indicate the hydrogen bonding with amino acids in the binding site (colored as red). The presented ligand was top-scored by the scoring function and obtained the lowest RMSD value (1.015 Å) after docking in AutoDock Vina.

**Figure 11 materials-14-06916-f011:**
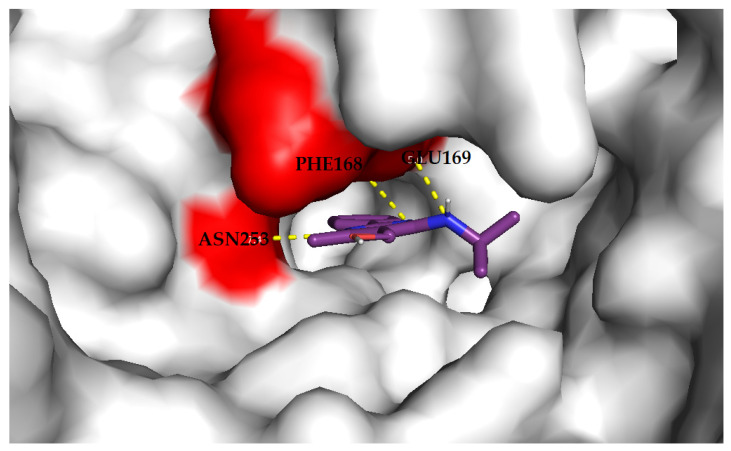
Compound **3h** in an adenosine A1 binding site. Yellow dashes indicate the hydrogen bonding with amino acids in the binding site (colored as red). The presented ligand was top-scored by the scoring function and obtained the lowest RMSD value (1.417 Å) after docking in AutoDock Vina.

**Table 1 materials-14-06916-t001:** Geometry of intra- and intermolecular hydrogen bonds for compounds **3c** and **3e**–**3h**.

Compounds	D–H···A	D–H	H···A	D···A	D–H···A
**3c**	O1–H1···N1 ^i^	0.84	2.01	2.825 (2)	162
	N4–H41···N3 ^ii^	0.88	2.31	3.151 (2)	159
	N4–H42···O1	0.88	2.29	2.834 (2)	120
**3e**	O1–H1···N1 ^i^	0.84	2.13	2.9525 (14)	165
	N4–H4···O1 ^ii^	0.88	2.48	3.1361 (14)	132
	N4–H4···O1	0.88	2.41	3.0116 (13)	126
**3f**	O1–H1···N1 ^i^	0.84	2.09	2.9199 (16)	172
	N4–H4···O1 ^ii^	0.88	2.13	2.9434 (16)	154
**3g**	O1–H1···N1 ^i^	0.84	2.08	2.9009 (12)	164
	N4–H4···O1 ^ii^	0.88	2.50	3.3210 (13)	155
	N4–H4···O1	0.88	2.55	3.0577 (14)	118
	C43–H431···O1 ^ii^	0.98	2.62	3.4038 (16)	138
**3h**	O1A–H1A···N1A ^i^	0.84	1.97	2.772 (5)	159
	N4A–H4A···O1A	0.88	2.18	2.767 (5)	124
	O1B–H1B···N1C ^ii^	0.84	2.01	2.78 (2)	152
	N4B–H4B···O1B	0.88	2.21	2.773 (16)	122
	O1C–H1C···N1B ^ii^	0.84	1.95	2.77 (2)	163
	N4C–H4C···O1C	0.88	2.20	2.78 (3)	123

Symmetry codes: **3c**: (i) x, −y + 3/2, z − 1/2; (ii) −x + 1, −y + 1, −z; **3e**: (i) −x, −y + 1, −z + 1; (ii) −x, −y + 1, −z; **3f**: (i) −x + 1, −y + 1, −z + 1; (ii) −x + 1, −y + 1, −z + 2; **3g**: (i) x, −y + 3/2, z − 1/2; (ii) −x, −y + 1, −z; **3h**: −x + 1/2, y, z + 1/2; −x, −y + 1, z − 1/2.

**Table 2 materials-14-06916-t002:** IC50 (µM) of tested compounds after 72 h incubation with cancerous cell lines, obtained in the neutral red uptake assay.

	HeLa	HepaRG	Caco-2	AGS	A172
**3a**	>250 µM	200.8 µM	>250 µM	209.2 µM	>250 µM
**3c**	>250 µM	>250 µM	>250 µM	>250 µM	>250 µM
**3d**	>250 µM	>250 µM	>250 µM	>250 µM	>250 µM
**3e**	>250 µM	>250 µM	>250 µM	>250 µM	>250 µM
**3f**	>250 µM	>250 µM	>250 µM	>250 µM	>250 µM
**3g**	209.4 µM	183.3 µM	>250 µM	>250 µM	146.7 µM
**3h**	>250 µM	132.3 µM	>250 µM	>250 µM	>250 µM
**3i**	>250 µM	>250 µM	>250 µM	>250 µM	>250 µM
Staurosporine	1 µM	1 µM	1 µM	1 µM	1 µM

**Table 3 materials-14-06916-t003:** MIC (µg mL^−1^) of compounds tested against a panel of microorganisms.

	3a	3c	3d	3e	3f	3g	3h	3i	Standard Agent (µg/mL)
*A. baumannii*	-	-	-	256	-	256	256	-	Levofloxacin: 0.5
*P. aeruginosa*	-	-	-	-	-	-	-	-	Levofloxacin: 1
*E. coli*	-	-	-	-		128	-	-	Gentamicin: 2
*E. faecalis*	-	-	-	256	-	-	-	-	Levofloxacin: 4
*K. pneumoniae*	-	-	-	-	-	-	-	-	Gentamicin: 2
*MRSA*	-	-	-	-	256	-	-	-	Levofloxacin: 1
*C. albicans*	256	-	-	-	-	-	-	-	Amphotericin B: 1

**Table 4 materials-14-06916-t004:** ADME analysis results.

	3a	3g	3h
Physiochemical properties	The compound has a molecular weight of 308.40 g/mol; number of heavy atoms and number of aromatic heavy atoms: 22 and 11, respectively; number of rotatable bonds: 4; number of H-bond acceptors and donors: 3 and 1, respectively. The value of the polar surface area (PSA) calculated using the topological polar surface area (TPSA), considering sulfur and phosphorus as polar atoms, is 71.32 Å^2^ [31].	The compound has a molecular weight of 257.33 g/mol; number of heavy atoms and number of aromatic heavy atoms: 19 and 12, respectively; number of rotatable bonds: 4; number of H-bond acceptors and donors: 3 and 2, respectively. The value of the polar surface area (PSA) calculated using the topological polar surface area (TPSA), considering sulfur and phosphorus as polar atoms, is 58.04 Å.	The compound has a molecular weight of 271.36 g/mol; number of heavy atoms and number of aromatic heavy atoms: 20 and 12, respectively; number of rotatable bonds: 4; number of H-bond acceptors and donors: 3 and 2, respectively. The value of the polar surface area (PSA) calculated using the topological polar surface area (TPSA), considering sulfur and phosphorus as polar atoms, is 58.04 Å.
Lipophilicity	The value partition coefficient between n-octanol and water (log Po/w) is 3.72 [32]. It is an average value of five freely available predictive models (i.e., XLOGP3 [33], WLOGP [34], MLOGP [35,36], SILICOS-IT [25], and iLOGP) [37].	The consensus value partition coefficient between n-octanol and water (log Po/w) is 2.68.	The consensus value partition coefficient between n-octanol and water (log Po/w) is 2.92.
Water solubility	Estimated by three predictors. The value of Log S (ESOL) [38] is −4.47, which makes a compound moderately soluble. The predicted value of solubility is 1.03∙10^−2^ mg/mL. The value of log S (Ali) [39] is −4.97, which also classifies the compound as moderately soluble. The value of solubility is 3.30 × 10^−3^ mg/mL. The value of log S (SILICOS-IT) [25] is −7.00, which classifies the compound as poorly soluble. The predicted value of solubility is 3.09 × 10^−5^ mg/mL.	The value of Log S (ESOL) is −3.56, which classifies a compound as soluble. The predicted value of solubility is 7.08 × 10^−2^ mg/mL. The value of log S (Ali) is −3.94, which also classifies the compound as soluble. The value of solubility is 2.99 × 10^−2^ mg/mL. The value of log S (SILICOS-IT) is −5.26, which classifies the compound as moderately soluble. The predicted value of solubility is 1.40 × 10^−3^ mg/mL.	The value of Log S (ESOL) is −3.74, which classifies a compound as soluble. The predicted value of solubility is 4.97 × 10^–2^ mg/mL. The value of log S (Ali) is −4.12, which classifies the compound as moderately soluble. The predicted value of solubility is 2.05 × 10^–2^ mg/mL. The value of log S (SILICOS-IT) is −5.65, which also classifies the compound as moderately soluble. The predicted value of solubility is 6.09 × 10^–4^ mg/mL.
Pharmacokinetics	One of the estimated predictors relates to skin permeability coefficient (Kp) [40]. The more negative Kp is, the less permeant a molecule is. The predicted Kp value of compound **3a** is −5.50 cm/s, the predicted interaction of a molecule with cytochromes P450 [41,42]. The inhibition of five isoforms, CYP1A2, CYP2C19, CYP2C9, CYP2D6, and CYP3A4, may cause pharmacokinetics-related drug–drug interactions. Compound **3a** is predicted to be an inhibitor of all of the five enzyme isoforms.	The predicted Kp value of compound **3g** is −5.70 cm/s. Compound **3g** is predicted to be an inhibitor of CYP1A2 and CYP2D6.	The predicted Kp value of compound **3h** is −5.66 cm/s. Compound **3g** is predicted to be an inhibitor of CYP1A2 and CYP2D6 and CYP3A4.
Druglikeness	Estimation of the chance to be an oral drug. The Swiss ADME software bases on five different predictors. Originally used by major pharmaceutical companies aiming to improve the quality of their chemical substances. The Lipinski (Pfizer) rule of five [43], Ghose (Amgen) [44], Veber (GSK) [45], Egan (Pharmacia) [46], and Muegge (Bayer) [47]. According to all of the predictors, compound **3a** is predicted to have a chance to be an oral drug.	According to all of the five predictors, compound **3g** is predicted to have a chance to be an oral drug.	According to all of the five predictors, compound **3h** is predicted to have a chance to be an oral drug.
Medicinal chemistry	Two complementary pattern recognition methods allow for the identification of potentially problematic fragments—assay interference compounds (PAINS) [48] and Brenk Structural alert [49]. Zero predicted alerts assist in creating a good druglike molecule.	Compound **3g** has zero predicted structural problematic fragments.	Compound **3h** has zero predicted structural problematic fragments.

**Table 5 materials-14-06916-t005:** List of predicted targets.

Compound	Target Class	Target Name	Protein Data Bank (PDB) Accession Code
**3a**	Phosphodiesterase	Phosphodiesterase 10A	5B4L
	Oxidoreductase	Cyclooxygenase-1	6Y3C
**3g**	Phosphodiesterase	Phosphodiesterase 4B	1ROR
	Family A G protein-coupled receptor	Adenosine A1 receptor	5UEN
**3h**	Family A G protein-coupled receptor	Adenosine A1 receptor	5UEN
	Family A G protein-coupled receptor	Adenosine A2a receptor	5IU4

## Data Availability

Data can be found in the article or Appendix A.

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
