# Peer review of "Synthesis, Crystal Structure, and Biological Evaluation of Novel 5-Hydroxymethylpyrimidines"

_materials, 2021, doi:10.3390/ma14226916_

Round 1

Reviewer 1 Report

The article “Synthesis, crystal structure and biological evaluation of novel 2 5-hydroxymethylpyrimidines” provides synthesis, crystal structure , anticancer and antimicrobial screening of ten 5-hydroxymethylpyrimidine derivatives.

The English of the manuscript requires serious attention, several inaccurate sentences   English must be checked by native English speaker

Information presented in many cases are not supported by References.

The literature citation is not consistent.

ADME of all derivatives must be provided and discussed

in silico analysis (docking and dynamics 50 ns) should be provided for the potent compounds to predict the target of action

conclusions need more elaboration

Author Response

“Extensive editing of English language and style required . The English of the manuscript requires serious attention, several inaccurate sentences   English must be checked by native English speaker”

The authors have performed the corrections by native speaker from professional language office. All changes have been added to the file. If Editor needs the original file from Native speaker please do not hesitate to ask us for it and we will send it directly to the Editor.

“The literature citation is not consistent”

The literature has been unified and prepared according to the publisher requirements.

“ADME of all derivatives must be provided and discussed; in silico analysis (docking and dynamics 50 ns) should be provided for the potent compounds to predict the target of action”

In silico analysis with ADME and target prediction have been added (see section 2.4 and 3.4)

“Conclusions need more elaboration”

The conclusions have been extended. Neverthless, looking on the similarity report authors noticed that the version where conclusions were added has not been evaluated by MDPI. So please check is the correct version with section: conclusions is added in MDPI file.

Reviewer 2 Report

The article “Synthesis, crystal structure and biological evaluation of novel  5-hydroxymethylpyrimidines” presents a series of 5-hydroxymethylpyrimidines that vary in 4 position by reduction of proper esters. The article is not worthy of being published in its actual form. The authors performed a single experiment without its duplication to validate the results). Compounds 3b and 3j show moderate toxicity: the IC50 is approximately 50uM. The concentration of seeded cells is much too high for the 96-well plate (105 cells). For these reasons, it is necessary to evaluate them in order to observe their efficiency.
The authors present a simple test (neutral red test) to demonstrate the antineoplastic properties of tested compounds possess. Additional tests are necessary to sustain this fact. 

Author Response

“The authors performed a single experiment without its duplication to validate the results)”
The neutral red uptake assay were performed based on established protocol (Repetto G. 2008, Narute). It is primary screening test, which can indicate activity of tested compound. It has been prepared in four replicates, from which arithmetic mean were taken. When results from each replicate were constant (standard deviation was also calculated), there was no point for further analysis. If results differed between replicates, additional test were performed for confirmation. Thus it was not a single experiment but the experiment flow was repeated. A specific trend was obtained.
Generally, when the group of potential active compounds are investigated, after screening tests when no sophisticated activity is observed any further analysis are performed regarding to the high costs and time consuming operations. Thus, you can see the papers where only one cytotoxicity test were performed (DOI: 10.1107/S2053229618012706) and you can also check the papers where the low IC50 results were obtained and here more detail analysis regarding molecular mechanisms were analysed (DOI: 10.3390/molecules26082296).
“Compounds 3b and 3j show moderate toxicity: the IC50 is approximately 50uM”
Compounds 3b and 3j showed cytotoxic effects against regular cell line RPTEC is human renal proximal tubule epithelial cells, non-cancerous cell line. Based on this results, compounds which shows high cytotoxicity are excluded from tests on cancerous cell lines, as, even if they were toxic for cancer, they could destroy kidneys, after administration to the patient.
“The concentration of seeded cells is much too high for the 96-well plate (105 cells). For these reasons, it is necessary to evaluate them in order to observe their efficiency”
The seeding density, as described in methods, was 10^5/ml which make 1-^4/well. We use this dilutions based on literature (e.g. Repetto G. 2008, Nature), material vendors recommendation (e.g. Thermofisher seeding density recommendation for 96-well plate is .x10^6/well) as well as on our experience with cell culture.
“The authors present a simple test (neutral red test) to demonstrate the antineoplastic properties of tested compounds possess. Additional tests are necessary to sustain this fact”
Neutral red uptake assay is one of the most widely use primary screening tests. It showed quantitative estimation of viable cells after treatment. It has been prepared in replicates to make sure and confirm the trend. It is sufficient test for wide screening. Some of the results were confirmed with cytotoxic effect (results were not shown). Further analysis, in our opinion are necessary for the compound showed high cytotoxic effects, as they can be potentially used as anticancer drugs. E.g. extended tests were made for other pyrimidine derivatives described in Stolarczyk M, et. Al. 2021, where IC50 for compound 3 were 50uM. With IC50 higher than 100 uM for all tested compounds, there was no basis to expand research which are expensive and time consuming

Reviewer 3 Report

The manuscript by Matera-Witkiewicz et al. describes the synthesis of some 5-hydroxymethylpyrimidines and their antimicrobial and antiproliferative effects. The compounds are only weakly active, however they are well caracterized by means of 1HNMR and IR spectroscopy, cristallography and mass spectrometry. Hence, in my opinion the manuscript deserves to be published in “materials” after the revisions listed below:

1) Reaxys database reports compound 2a as commercially available, this information should be reported in the paper.

2) Compound 2c is reported in the following papers, at least one of these should be cited:

1) Indian Journal of Chemistry - Section B Organic and Medicinal Chemistry, 1996, vol. 35, p. 598 - 601

2) Bulletin of the Academy of Sciences of the USSR Division of Chemical Science, 1990, vol. 39, # 1.2, p. 130 - 134

3) A standard agent should be added in both Table 2 and Table 3 in order to compare IC50 and MIC values respectively.

Round 2

Reviewer 1 Report

manuscript has been improved

Author Response

The authors would like to thank for the revision and suggestions which improved the manuscript. We are thankfull for acceptance all corrections.

Reviewer 2 Report

The title of the article is: “Synthesis, crystal structure and biological evaluation of novel 5-hydroxymethylpyrimidines” and the aims: “we present the results of the cytotoxic and antimicrobial study of a series of 5-hydroxymethylpyrimidines that vary in the 4-position and comparison of their structural properties and biological activity.” This means that the structural properties and biological activity of all synthesized compounds will be presented in the article, which is not the case of this article. However, despite the good and adequate characterization of the compound from the chemical point of view, the biological evaluation assessment is not to the desired standards for publication in the journal.

Minor observation

The authors have to introduce the catalogue number for each used cell culture (e.g. AGS CRL-1739), or the laboratory that provided them.

The method of IC50 calculations was not provided.

Author Response

The authors would like to thank for the revision. Regarding to the comment:

"This means that the structural properties and biological activity of all synthesized compounds will be presented in the article, which is not the case of this article. However, despite the good and adequate characterization of the compound from the chemical point of view, the biological evaluation assessment is not to the desired standards for publication in the journal"

We would like to explain that according to the protocols for screening analysis (Repetto Nature 2008) the experiment was done for all compounds (see section with biological assays) neverthless some of the compounds have presented toxicity for regular cell lines thus they must been excluded from further experiments with cancer cell lines. It has been pointed in the manuscript.

Regarding to the comment:

"The authors have to introduce the catalogue number for each used cell culture (e.g. AGS CRL-1739), or the laboratory that provided them."

Following cell lines have been used:

RPTEC MTOX1030
HeLa 93021013
CaCo-2 86010202
HepaRG MTOX1010
AGS 89090402
A172 88062428 

Reagarding to the comment: "The method of IC50 calculations was not provided."

The method is described in details in the Repetto Nature protocol: "Draw dose-response curves and, where possible, calculate the concentration of a test chemical reflecting a 50% inhibition of the uptake (IC50) and the confidence interval using a methematical model, for example, a Hill function or logistic regression. Alternatively, graphical fitting methods can be employed."

The authors decides not to put it in the manuscript to not extend the text. The same approach has been used in others part of manuscript, where all protocol is not rewritten just the reference is added to the text. Nevertheless, if it seems to be indispencible the authors delcare to put it in the text.